# Coupled Dynamic Analysis and Decoupling Optimization Method of the Laser Gyro Inertial Measurement Unit

**DOI:** 10.3390/s20010111

**Published:** 2019-12-23

**Authors:** Fang Fang, Wenhui Zeng, Zilong Li

**Affiliations:** 1Wuhan National Laboratory for Optoelectronics, Huazhong Institute of Electro-Optics, Wuhan 430223, China; fangrifang@gmail.com (F.F.); lizilong_st@126.com (Z.L.); 2School of Electromechanical and Architectural Engineering, Jianghan University, Wuhan 430010, China

**Keywords:** ring laser gyro (RLG), inertial measurement unit (IMU), mechanical dither, coupled vibration, multi-rigid body dynamics

## Abstract

The mechanical dithered ring laser gyro (RLG) effectively overcomes the lock-in effect and ensures the sensitive accuracy of the low angular rate for the gyro. However, in the inertial measurement unit (IMU) system, the dither excitation of three RLGs causes the coupled vibration of the IMU structure, which could seriously limit the measuring accuracy of RLGs. In this paper, the vibration frequency response characteristic of laser gyro IMU is taken as the focus point, and the method of multi-rigid body dynamics is used to establish the dynamic model of IMU suitable for vibration frequency response analysis. On the basis of the model, the multi-degree-of-freedom coupling vibration of IMU with the gyro dither excitation is clearly described. A new IMU dynamic decoupling optimization method is proposed to minimize the coupled vibration frequency response, and compared with the previous optimal design method. The prototype experimental test results show that the coupled vibration of IMU is restrained more effectively by the proposed new method than by the previous optimal design method. Finally, on the basis of this new method, the measuring accuracy of the RLGs in the IMU system is improved, which is quite useful for practical engineering application.

## 1. Introduction

Gyroscopes are the key angular rate sensors. In the current applications of inertial navigation, two types of gyroscopes dominate, that is, the optoelectronic types including ring laser gyros (RLGs) [1,2] and fiber optic gyros (FOGs) [3,4], based on the Sagnac effect, and solid vibratory types such as hemispherical resonator gyros (HRGs) [5,6], based on the Coriolis force. Among these products, the RLG is the main inertial device owing to the high performance and reliabilities to vibration and temperature, especially for the harsh environments. The FOG senses the angular rate by phase shift between two counter-propagating waves in a fiber coil, and its performance is up to a similar grade to that of the RLG, but it may be more easily affected by external dynamic, thermal, or stress perturbations [7]. The HRG detects the precession angle of the stress wave in a quartz hemispherical resonator so as to sense the angular rate [8]. Compared with the optoelectronic gyros, the HRG has superiority for its high reliability, low power, and good environmental adaptability. On the other hand, the chip-scale optoelectronic gyroscopes will play important roles in the gyroscope market owing to the low costs and miniaturization [9]. The resonant micro-optic gyro (RMOG) is a promising sensor in the field of integrated optical gyros, and more innovative studies will focus on RMOG’s key technology, that is, the silica-on-silicon ring resonator with an ultra-high Q-factor and the monolithic integration based on gyro on a chip [10,11].

Even though more and more types of high-performance gyroscopes are being studied and manufactured, the RLG still plays an extremely important role in the field of high precision inertial navigation owing to its mature manufacturing process. In the family of RLGs, the mechanical dithered RLG is most widely applied, which overcomes the gyro’s lock-in effect by the dither bias system and ensures the high measurement accuracy of the gyro for a low angular rate [12].

However, the mechanical dither of RLG becomes the internal vibration source in the inertial navigation system (INS), especially for the inertial measurement unit (IMU), which is the core component of the INS. Even if the RLGs are orthogonally mounted on the support structure in the IMU, the spatial multi-degree-of-freedom (DOF) coupled dynamic behavior with the excitation of each gyro could still be caused by the asymmetry of the IMU structure, which could seriously affect the accuracy of the gyros in IMU and greatly restrict the performance of INS [13].

In order to eliminate the RLG’s lock-in effect so as to ensure the dither dynamics, many studies have been carried out in the areas of RLG’s dither control methods and the lock-in error compensation technology. In terms of dither control, Wang [14] deeply studied the influence of the dither frequency bias on the reduction of dynamic lock-in width and the accuracy of the gyro. Fan [15] accurately extracted the dither frequency bias through the sensitive angular rate of the gyro, and used it to improve the stability of dither control. In terms of lock-in error compensation, Song [16] proposed a new algorithm to estimate the Sagnac phase errors due to lock-ins, and to compensate the random walk error of the RLG. Subsequently, Fan [17] proposed a novel lock-in error correction method to pick up the lost information and remove the random walk error effectively. The above studies have offered important technical support for the design of the dither bias system, and meanwhile laid the foundations for the structure design of the laser gyro IMU.

The studies of the structural dynamics on the laser gyro IMU can be divided into two categories, specifically, the studies on the calibration of the systematic errors of RLGs in the INS, and the studies on the optimal design of the IMU structures. In terms of the former, the studies are almost focused on the calibration and compensation of the attitude errors of the RLGs in IMU, that is, Zheng [18] employed a multi-position calibration method to estimate the g-dependent bias of the RLG in the dual-axis INS; Kim and Wang [19,20] proposed the calibration technique of gyro sensitive-axis deviation angle under static and linear vibration conditions, respectively; and, recently, Tu [21] accurately analyzed the time-domain response of the gyro deviation angle in a biaxial INS through modal decoupling of the IMU.

In terms of the optimal design of the laser gyro IMU structure, a dynamic-vibration-absorber-type solution was implemented on the inertial sensor assembly by Jamil [22] to improve the response of the IMU structure to the RLG’s dither. Subsequently, many researchers focused on the IMU dither decoupling, and proposed the relevant optimal design principles [23,24,25] such as the design requirements of the overlap between the IMU centroid and the center of the multiple-damper layout, and that between the gyro sensitive axis and the principle inertia axis of IMU.

Although the design principles [23,24,25] are proposed, they are sometimes difficult to be satisfied in practical engineering owing to various constraints such as the size of IMU and the installation mode of printed circuit board (PCB) in actual IMU or INS design. Secondly, these design principles are not available for some new structures of IMU, that is, the skewed redundant INS [26], as the orthogonal layouts of the RLGs in the new types of IMU are always impossible. Furthermore, whatever the studies on the calibration or on the structure optimal design of the laser gyro IMU, the methods of modelling and optimization applied are always based on the traditional Newton–Euler method, and mainly focus on the time domain response of IMU vibration, which is not available for frequency decoupling.

Therefore, frequency-domain-analysis based modelling and optimization methods [27] may become the new key techniques for the vibration decoupling and optimization of the laser gyro IMU. Frequency-domain analytical methods focus on the analysis and optimization of the modal and frequency response for the mechatronic systems, which have been widely used in multiple precision manufacturing fields, that is, the dynamic analysis and optimization of the high-performance machine tools [28], the design of the active vibration isolators for the precision instruments [29], and the NVH (noise, vibration, and harshness) analysis of the vehicle [30], among others.

In this paper, the vibration frequency response characteristics of the laser gyro IMU are taken as the focus point, and the multi-rigid body dynamic model of IMU for the frequency response analysis is established to clearly describe the multi-DOF coupled vibration behavior in IMU under gyro dither excitation. Then, the IMU dynamic decoupling optimization method is proposed to minimize the coupled vibration response of IMU, which overcomes the limitation of the previous IMU optimal design principle constrained by the actual design conditions. The results of the prototype experiment show that the dynamic decoupling optimization method of IMU proposed in this paper could effectively suppress the coupled vibration in IMU and improve the measurement accuracy of the RLGs in the IMU system.

The remainder of this paper is organized as follows. The description of the structural components of laser gyro IMU is presented in Section 2. The dynamic model of laser gyro IMU is constructed and its analysis results are given in Section 3. The optimization design of IMU dynamic decoupling is given in Section 4. The experimental verification of IMU dynamic decoupling is presented in Section 5, and Section 6 provides the conclusion.

## 2. Structure Components of Laser Gyro IMU

IMU is the core component of the INS, which obtains the navigation information such as speed, position, and attitude of the carrier in real-time through processing and solving the inertial signals provided by the three-axis gyro and accelerometer [31]. A typical laser gyro IMU consists of a sensor supporting structure; an outer housing; vibration isolators (also called dampers); a group of orthogonally distributed measuring devices, such as three RLGs; and accelerometers. The RLGs and the accelerometers are mounted on the sensor supporting structure, and multiple vibration isolators connect the outer housing and the sensor supporting structure. Considering the bearing capacity of a single rubber damper, the IMU always uses an eight-point damping method, that is, the outer housing and the sensor supporting structure are connected by eight vibration isolators [32]. The structure diagram of the laser gyro IMU assembly is shown in Figure 1.

## 3. Dynamic Modelling and Discussion of Laser Gyro IMU

### 3.1. Multi-Rigid Body Dynamic Modeling of Laser Gyro IMU

Generally, the mechanical dither frequency of RLG is much lower than the first-order modal frequency of the sensor supporting structure in IMU. Therefore, the sensor supporting structure and the resonance cavity of RLG can be simplified into rigid bodies, and the dithering wheel is equivalent to a torsion spring. In the work of [33], dynamic structures are simplified as multiple rigid bodies, springs, and hinges, and a topological diagram is used to represent the constraint relations of the dynamic model. On that basis, in this paper, the multi-rigid body dynamic model of the IMU structure is developed based on the method of topology analysis of constraint relations. The three elements, such as spring-damper units, multiple rigid bodies, and ideal constrained pairs, are combined with the topology form of constraint relations of IMU to describe the RLG’s dither and multi-point vibration isolation in IMU.

The modeling process is divided into four steps.


**Step 1: Definitions of the basic elements of the multi-rigid body system of IMU**


The vibration displacements of the centroid of single rigid body relative to the global coordinate system are defined as the generalized coordinates of the dynamic equation: $q_{i}={\left[\begin{array}{cccccc} x_{i} & y_{i} & z_{i} & \alpha_{i} & \beta_{i} & \gamma_{i} \end{array}\right]}^{T}$ (i is the serial number of bodies in the system). The generalized coordinates $\left(\begin{array}{ccc} \alpha & \beta & \gamma \end{array}\right)$ set as small angular displacements, which can be described by cardan angles.

The schematic diagram of the IMU dynamic model is shown in Figure 2, and the global coordinate system is defined as $\left\{\sum_{g}-xyz\right\}$, the dither coordinate systems of the three RLGs are $\left\{\sum_{r_{i}}-xyz\right\}\;$(i=1~3), and the local coordinate systems of eight vibration isolators are $\left\{\sum_{v_{j}}-xyz\right\}\;$(i=1~8). In this paper, the dynamic equation of IMU is described in the global coordinate system, which is fixed on the ground. The dither coordinate systems of RLGs and the coordinate systems of vibration isolators are fixed on the corresponding conjoined bases (i.e., three gyro cavities or eight vibration isolators), and they will change with the movements of the conjoined bases.

As shown in Figure 2, the whole IMU dynamic model includes four motion units, such as the sensor supporting structure and three gyro cavities. Without considering the constraint relationship, the model contains 6 × 4 = 24 DOFs. Further, the rotation pair between the gyro cavity and the sensor support structure is defined to limit the other five DOFs between the IMU and each RLG, except the direction of dither. Whereby, the DOFs of model are finally reduced to nine, so the dynamic model is composed of a nine-order differential equation.


**Step 2: Dynamic modeling of system without considering constraints**


The serial numbers of the sensor supporting structure and the three gyro resonators are sequentially from 1 to 4. The centroid coordinate system of the sensor supporting structure coincides with the global coordinate system. The direction cosine matrix from the global coordinate system $\left\{\Sigma_{g}\right\}$ to the three gyro local coordinate systems $\left\{\Sigma_{r_{i}}\right\}$ is defined as $A_{i}^{gr}$ (*i* = 2~4), then the system mass matrix in the global coordinate system $\left\{\Sigma_{g}\right\}$ can be written as follows:(1)$$M=\left[\begin{array}{ccc} M_{1}^{g} & & \\ & \begin{array}{cc} M_{2}^{g} & \\ & M_{3}^{g} \end{array} & \\ & & M_{4}^{g} \end{array}\right],$$
where $M_{i}^{g}=\left[\begin{array}{cc} {\left(A_{i}^{gr}\right)}^{T} & 0\\ 0 & {\left(A_{i}^{gr}\right)}^{T} \end{array}\right]M_{i}^{r}\left[\begin{array}{cc} A_{i}^{gr} & 0\\ 0 & A_{i}^{gr} \end{array}\right]$, *i* = 2, 3, 4.

The stiffness matrices of eight vibration isolators in each of the local coordinate systems $\left\{\Sigma_{v_{j}}\right\}$ can be written as follows:(2)$$K_{vj}=diag\left(\begin{array}{cccccc} k_{p} & k_{r} & k_{r} & 0 & 0 & 0 \end{array}\right),j=1\sim 8.$$

Equation (2) considers the linear stiffness $\left(\begin{array}{ccc} k_{p} & k_{r} & k_{r} \end{array}\right)$ of the vibration isolator along each of the three coordinate axes. The structure of the vibration isolator in IMU is given in Figure 3, and the stiffness parameters in the three axes can be checked according to the product manual provided.

Correspondingly, the stiffness matrices of three gyro resonators in each local coordinate system $\left\{\Sigma_{r_{i}}\right\}$ can be written as follows:(3)$$K_{ri}=diag\left(\begin{array}{cccccc} k_{xi} & k_{yi} & k_{zi} & k_{rxi} & k_{ryi} & k_{d} \end{array}\right),i=2\sim 4,$$
where the dither stiffness of RLG is $k_{d}$.

On the basis of the Lagrange method, without constraints, the dynamic equation of IMU can be derived as follows:(4)$$M\ddot{q}+Kq=0,$$
where ***M*** is represented in Equation (1), and ***K*** is as follows:(5)$$K=\left[\begin{array}{cccc} K_{11} & -K_{12} & -K_{13} & -K_{14}\\ -K_{21} & K_{22} & 0 & 0\\ -K_{31} & 0 & K_{33} & 0\\ -K_{41} & 0 & 0 & K_{44} \end{array}\right],$$
where
(6)$$K_{11}=\overset{8}{\underset{j=1}{\sum }}{\left(T_{j}^{gv}\right)}^{T}{\left(R_{j}^{gv}\right)}^{T}K_{vi}R_{j}^{gv}T_{j}^{gv}+\overset{4}{\underset{i=2}{\sum }}{\left(T_{i}^{gr}\right)}^{T}{\left(R_{i}^{gr}\right)}^{T}K_{ri}R_{i}^{gr}T_{i}^{gr},$$
(7)$$K_{1i}={\left(T_{i}^{gr}\right)}^{T}{\left(R_{i}^{gr}\right)}^{T}K_{ri}R_{i}^{gr}I,\;i=2\sim 4,$$
(8)$$K_{ii}=I{\left(R_{i}^{gr}\right)}^{T}K_{ri}R_{i}^{gr}I,\;i=2\sim 4,$$
(9)$$K_{i1}=I{\left(R_{i}^{gr}\right)}^{T}K_{ri}R_{i}^{gr}T_{i}^{gr},\;i=2\sim 4.$$

In Equations (6)–(9), $T_{j}^{gv}\;$(*j* = 1~8) and $T_{i}^{gr}\;$(*i* = 2~4) are the vibration displacement transfer matrix from the IMU centroid to the center of each vibration isolator and to the dither center of each RLG, respectively. $R_{j}^{gv}\;$(*j* = 1~8) and $R_{i}^{gr}\;$(*i* = 2~4) are the pose transformation matrix from the global coordinate system $\left\{\Sigma_{g}\right\}$ to the local coordinate system $\left\{\Sigma_{v}\right\}$ of each vibration isolator and to the local coordinate system $\left\{\Sigma_{r}\right\}$ of each RLG’s resonator, respectively.


**Step 3: Topology analysis of the constraint relation**


An additional revolute pair is applied between each RLG’s resonant cavity and the sensor supporting structure. Eight vibration isolators connecting the sensor supporting structure and the ground are equivalent to spring-dampers. The constraint topological form of IMU is shown in Figure 4, in which the serial number 0 is the ground, 1 is the sensor supporting structure of IMU, and 2~4 are RLG’s resonant cavities.

The joint matrix ***H*** and the variable separation matrix ***D*** of each revolute pair are defined, respectively, as follows:(10)$$H=diag\left(\begin{array}{cccccc} 1 & 1 & 1 & 1 & 1 & 0 \end{array}\right),$$
(11)$$D=\left[\begin{array}{cccccc} 0 & 0 & 0 & 0 & 0 & 1 \end{array}\right].$$

Considering the revolute constraints between the RLGs and IMU, the established constraint matrix of IMU is as follows:(12)$$B=\left[\begin{array}{cccc} I_{1} & 0 & 0 & 0\\ P_{12} & Q_{12} & 0 & 0\\ P_{13} & 0 & Q_{13} & 0\\ P_{14} & 0 & 0 & Q_{14} \end{array}\right]\left[\begin{array}{cc} I_{1} & \\ & \begin{array}{cc} D & \\ & \begin{array}{cc} D & \\ & D \end{array} \end{array} \end{array}\right]=\left[\begin{array}{cccc} I_{1} & 0 & 0 & 0\\ P_{12} & Q_{12}D & 0 & 0\\ P_{13} & 0 & Q_{13}D & 0\\ P_{14} & 0 & 0 & Q_{14}D \end{array}\right],$$
where
(13)$$P_{1i}=I{\left(R_{i}^{gr}\right)}^{T}HR_{i}^{gr}T_{i}^{gr},\;i=2\sim 4,$$
(14)$$Q_{1i}=I{\left(R_{i}^{gr}\right)}^{T}\left(I-H\right),\;i=2\sim 4.$$


**Step 4: Dynamic modeling of the multi-rigid body system of IMU**


Substituting the constraint matrix (12)–(14) into the dynamic Equation (4) of the non-constraint system, and considering the hysteresis damping characteristics of the rubber damper [34] (i.e., the elastic and damping effects of the vibration isolator are F=(k+iωc)x=k(1+iβ)x, where β is the hysteresis damping coefficient), meanwhile considering the quality factor Q of RLG’s dither system, so the dynamic model of IMU can be obtained as follows:(15)$$\hat{M}\ddot{q}+\hat{K}\left(1+i\beta \right)q=0,$$
(16)$$\hat{K}=\left[\begin{array}{cc} {\hat{K}}_{11} & {\hat{K}}_{12}\\ {\hat{K}}_{21} & {\hat{K}}_{22} \end{array}\right],$$
(17)$${\hat{K}}_{11}=\overset{8}{\underset{i=1}{\sum }}\left[\begin{array}{cccccc} k_{r} & 0 & 0 & 0 & k_{r}z_{i} & -k_{r}y_{i}\\ 0 & k_{p} & 0 & -k_{p}z_{i} & 0 & k_{p}x_{i}\\ 0 & 0 & k_{r} & k_{r}y_{i} & -k_{r}x_{i} & 0\\ 0 & -k_{p}z_{i} & k_{r}y_{i} & k_{p}z_{i}{}^{2}+k_{r}y_{i}{}^{2}+\frac{k_{d_{1}}}{8} & -k_{r}x_{i}y_{i} & -k_{p}x_{i}z_{i}\\ k_{r}z_{i} & 0 & -k_{r}x_{i} & -k_{r}x_{i}y_{i} & k_{r}z_{i}{}^{2}+k_{r}x_{i}{}^{2}+\frac{k_{d_{2}}}{8} & -k_{r}z_{i}y_{i}\\ -k_{r}y_{i} & k_{p}x_{i} & 0 & -k_{p}x_{i}z_{i} & -k_{r}z_{i}y_{i} & k_{p}x_{i}{}^{2}+k_{r}y_{i}{}^{2}+\frac{k_{d_{3}}}{8} \end{array}\right],$$
(18)$${\hat{K}}_{12}=\left[\begin{array}{ccc} 0 & 0 & 0\\ 0 & 0 & 0\\ 0 & 0 & 0\\ -k_{d_{1}} & 0 & 0\\ 0 & k_{d_{2}} & 0\\ 0 & 0 & -k_{d_{3}} \end{array}\right],$$
(19)$${\hat{K}}_{21}=\left[\begin{array}{cccccc} 0 & 0 & 0 & -k_{d_{1}} & 0 & 0\\ 0 & 0 & 0 & 0 & k_{d_{2}} & 0\\ 0 & 0 & 0 & 0 & 0 & -k_{d_{3}} \end{array}\right],$$
(20)$${\hat{K}}_{22}=\left[\begin{array}{ccc} k_{d_{1}} & 0 & 0\\ 0 & k_{d_{2}} & 0\\ 0 & 0 & k_{d_{3}} \end{array}\right],$$
(21)$$\hat{M}=\left[\begin{array}{cc} {\hat{M}}_{11} & 0\\ 0 & {\hat{M}}_{22} \end{array}\right],$$
(22)$$\begin{array}{c} {\hat{M}}_{11}=[\begin{array}{cccc} 3m+m_{b} & & & \\ & 3m+m_{b} & & -m(z_{r2}+z_{r3}+z_{r4})\\ & & 3m+m_{b} & m(y_{r2}+y_{r3}+y_{r4})\\ & -m(z_{r2}+z_{r3}+z_{r4}) & m(y_{r2}+y_{r3}+y_{r4}) & m(y_{r2}^{2}+y_{r3}^{2}+y_{r4}^{2}+z_{r2}^{2}+z_{r3}^{2}+z_{r4}^{2})+2I_{x}+I_{xx}\\ m(z_{r2}+z_{r3}+z_{r4}) & & -m(x_{r2}+x_{r3}+x_{r4}) & I_{xy}-m(x_{r2}y_{r2}+x_{r3}y_{r3}+x_{r4}y_{r4})\\ -m(y_{r2}+y_{r3}+y_{r4}) & m(x_{r2}+x_{r3}+x_{r4}) & & I_{xz}-m(x_{r2}z_{r2}+x_{r3}z_{r3}+x_{r4}z_{r4}) \end{array}\\ \begin{array}{cc} m(z_{r2}+z_{r3}+z_{r4}) & -m(y_{r2}+y_{r3}+y_{r4})\\ & m(x_{r2}+x_{r3}+x_{r4})\\ -m(x_{r2}+x_{r3}+x_{r4}) & \\ I_{xy}-m(x_{r2}y_{r2}+x_{r3}y_{r3}+x_{r4}y_{r4}) & I_{xz}-m(x_{r2}z_{r2}+x_{r3}z_{r3}+x_{r4}z_{r4})\\ m(x_{r2}^{2}+x_{r3}^{2}+x_{r4}^{2}+z_{r2}^{2}+z_{r3}^{2}+z_{r4}^{2})+I_{x}+I_{y}+I_{yy} & I_{yz}-m(y_{r2}z_{r2}+y_{r3}z_{r3}+y_{r4}z_{r4})\\ I_{yz}-m(y_{r2}z_{r2}+y_{r3}z_{r3}+y_{r4}z_{r4}) & m(x_{r2}^{2}+x_{r3}^{2}+x_{r4}^{2}+y_{r2}^{2}+y_{r3}^{2}+y_{r4}^{2})+2I_{y}+I_{zz} \end{array}], \end{array}$$
(23)$${\hat{M}}_{22}=\left[\begin{array}{ccc} I_{Z} & & \\ & I_{Z} & \\ & & I_{Z} \end{array}\right].$$

In Equations (16)–(23), $\left(\begin{array}{ccc} x_{i} & y_{i} & z_{i} \end{array}\right)\;$(*i* = 1~8) are the positions of eight vibration isolators relative to the centroid of IMU, and $\left(\begin{array}{ccc} x_{ri} & y_{ri} & z_{ri} \end{array}\right)\;$(*i* = 2~4) are the positions of three RLG’s cavities relative to the centroid of IMU. It should be emphasized that this paper focuses on the frequency response analysis of the coupled vibration behavior of three RLGs in IMU. Therefore, the external generalized force is not considered in the above dynamic model.

### 3.2. Discussion of Coupled Vibration Behavior of IMU

Through analyzing the stiffness matrix $\hat{K}$ in Equations (15)–(23), we can find that the stiffness distribution of IMU is related to the positions $\left(x_{i}\;y_{i}\;z_{i}\right)$ of the vibration isolators relative to the IMU centroid (*i* = 1~8), besides the stiffness $\left(\begin{array}{ccc} k_{p} & k_{r} & k_{r} \end{array}\right)$ of the vibration isolators in three axes and the torsional stiffness $\left(\begin{array}{ccc} k_{d_{1}} & k_{d_{2}} & k_{d_{3}} \end{array}\right)$ of three RLGs in each dither direction.

Similarly, the mass distribution of the system is related to the mass properties of the sensor supporting structure (including the mass parameter $m_{b}$ and inertia parameters $\left[I_{xx},I_{yy},I_{zz},I_{xy},I_{yz},I_{zx}\right]$), the inertia $I_{z}$ of each RLG along the respective dither directions, and the positions $\left(x_{ri}\;y_{ri}\;z_{ri}\right)$ (*i* = 2~4) of three RLGs relative to the centroid of the IMU.

The discussion of the coupled vibration behavior of the IMU under gyro dither is equivalent to the diagonalizing study of the mass matrix $\hat{M}$ and stiffness matrix $\hat{K}$ in the dynamic Equations (15)–(23). Therefore, the conditions for vibration decoupling of IMU can be derived as follows:(1)The IMU centroid coincides with the equivalent elastic center of the vibration isolators, that is, the eight vibration isolators, are symmetrically distributed with respect to the IMU. At this time, the stiffness matrix $\hat{K}$ in the model is diagonalized. The specific condition is as follows:(24)$$\overset{8}{\underset{i=1}{\sum }}x_{i}=\overset{8}{\underset{i=1}{\sum }}y_{i}=\overset{8}{\underset{i=1}{\sum }}z_{i}=\overset{8}{\underset{i=1}{\sum }}x_{i}y_{i}=\overset{8}{\underset{i=1}{\sum }}x_{i}z_{i}=\overset{8}{\underset{i=1}{\sum }}y_{i}z_{i}=0.$$(2)The principle inertia axes in the IMU centroid coordinate system are respectively along the dither axes of each RLG, that is, the non-diagonal terms in the mass matrix are zero. At this time, the inertia matrix $\hat{M}$ in the model is diagonalized. The specific condition is as follows:(25)$$\begin{array}{cc} I_{xy}-m(x_{r_{2}}y_{r_{2}} & +x_{r_{3}}y_{r_{3}}+x_{r_{4}}y_{r_{4}})=I_{xz}-m\left(x_{r_{2}}z_{r_{2}}+x_{r_{3}}z_{r_{3}}+x_{r_{4}}z_{r_{4}}\right)\\ & =I_{yz}-m\left(y_{r_{2}}z_{r_{2}}+y_{r_{3}}z_{r_{3}}+y_{r_{4}}z_{r_{4}}\right)=0. \end{array}$$


The similar decoupling studies can be found in the literatures [23,24,25,26], such as Equations (16)–(18) in the work of [23], Section 3.1 in the work of [25], and Section 4 in the work of [26]. Among the above previous studies, the work of [26] is the relatively recent study on the optimal design of the laser gyro IMU, where a new skewed redundant IMU by employing four RLGs is designed, and the dithered coupling of RLGs in the redundant configuration of IMU is considered. In the work of [26], the methods of increasing the weight of the IMU and allocating the RLGs’ frequencies reasonably are applied to reduce the coupled vibration, but no analytical model representing the coupled dynamics of IMU and relevant decoupling optimization method are involved.

Unlike the work of [26], other papers [23,24,25] propose similar decoupling conditions (i.e., the inertia product in the mass matrix is 0, namely $I_{xy}=I_{xz}=I_{yz}=0$) as Equations (24) and (25) in this paper.

By comparing the IMU dynamic decoupling conditions in Equations (24) and (25) and the previous decoupling conditions in the works of [23,24,25], some conclusions could be derived as follows:(1)The proposed decoupling condition such as Equation (25) is more general, compared with those in the works of [23,24,25]. The reason is that, in the actual IMU structure, the distance from the centroid of the IMU to each RLG is always not 0 (i.e., $\;x_{ri}\neq 0$, $\;y_{ri}\neq 0$, $\;z_{ri}\neq 0$), in that case, the dynamic decoupling is not achieved just by satisfying the equation $I_{xy}=I_{xz}=I_{yz}=0$.(2)On the other hand, the condition of $I_{xy}=I_{xz}=I_{yz}=0$ means a fully symmetrical structure of IMU, which is also not easy to be satisfied in the actual design of INS owing to other design requirements (such as mass, size, and layouts of PCBs or sensors).


In summary, no matter which decoupling conditions are considered, such as Equations (24)–(25) in this paper or the similar ones in the works of [23,24,25], they are all based on the diagonalizing of the system stiffness and the mass matrices. However, in order to obtain the minimum-coupled vibration response of the IMU, as well as improve the engineering availability, it is still necessary to analyze the vibration response in the frequency domain, so as to propose a new optimal design method for the IMU structure, which will be focused in the next section in this paper.

## 4. Optimization Design of IMU Dynamic Decoupling

On the basis of the established IMU dynamic model in Section 3, the IMU coupling frequency response amplitude can be minimized by optimizing the IMU dynamic parameters. The frequency response function matrix under vibration excitation is obtained as $H\left(\omega \right)={\left[\left(I+jG\right)K-\omega^{2}M\right]}^{-1}$ [35]. Extracting the *l*th row and *p*th column of H(ω), the following can be obtained:(26)$$H_{lp}\left(\omega \right)=\frac{x_{l}\left(\omega \right)}{f_{p}\left(\omega \right)}=\overset{N}{\underset{r=1}{\sum }}\frac{\psi_{lr}\psi_{pr}}{M_{r}\left[\frac{\left(1+jg\right)K_{r}}{M_{r}}-\omega^{2}\right]}.$$

Equation (26) represents that, if the point p is excited, the frequency response function of point l is obtained. If point *p* and point l are in different directions, Equation (26) represents the coupling frequency response of the system. At the same time, the formula also describes the mapping relationship between the structural parameters (positional relationship of vibration isolator relative to IMU centroid, the parameters of stiffness or damping, and so on) and dynamic response of the system.

According to Equation (26), with the aid of an intelligent algorithm, such as genetic algorithm, the coupling frequency response in IMU is regarded as the optimized objective, and the positions of vibration isolators relative to the IMU centroid ($\left(x_{i},y_{i},z_{i}\right)$ in Equation (17)) are the optimized parameters to achieve the IMU dynamic decoupling.

### 4.1. Initial Parameters

The IMU structure before optimization and the serial number of RLGs and vibration isolators are shown in Figure 5.

In Figure 5, the sensor supporting structure of IMU is designed beforehand according to the decoupling principles of Equations (24) and (25), in order to make sure that the off-diagonal terms in the IMU inertia properties are smaller than the 1/50 of diagonal terms. The vibration isolators are chosen as AM003-15 rubber dampers manufactured by Corp. Lord. The other initial physical properties of the IMU structure are as follows:(1)Weight of sensor supporting structure: 9.1 kg;(2)Dither frequencies of RLGs: no. ①–③ are 490 Hz, 570 Hz, and 670 Hz, respectively;(3)Quality factor of the gyro: 150;(4)Stiffness of vibration isolator: axial 58 N/mm, radial 65 N/mm;(5)Damping coefficient of vibration isolator: 0.2;(6)Positions of the vibration isolators relative to the IMU centroid: as shown in Table 1.

### 4.2. Design Parameters

According to Equations (15)–(23), the structure parameters to determine the IMU dynamic response include the following: (1) weight of sensor supporting structure; (2) inertia distribution of the sensor supporting structure around its centroid; (3) the axial and radial stiffness of the vibration isolators; and (4) the positions and poses of the vibration isolators relative to the IMU centroid coordinate system.

Owing to the limitation of design factors, such as system layout, size, and installation mode, in this paper, the positions of eight vibration isolators relative to the IMU centroid are regarded as the design parameters. For the further sake of engineering achievability, only the *x* and *z* coordinates of each of vibration isolators in IMU centroid coordinate system are optimized. Specifically, the positions of no. 1~4 vibration isolators can be adjusted along the *x* and *z* axes of the IMU centroid coordinate system, and the positions of no. 5~8 vibration isolators can only be adjusted along the *z* axis, as shown in Figure 6.

Figure 6 is along the *y*-axis of the centroid coordinate system, where the positions of no. 2, 4, 6, and 8 vibration isolators overlap the positions of no. 1, 3, 5, and 7 vibration isolators, respectively. The design parameters in Figure 6 are $x_{i}\left(i=1\sim 4\right)$, $z_{i}\left(i=1\sim 8\right)$. The initial values of these design parameters are given in Table 1. The range of these parameters are as follows:(1)$x_{i}\epsilon \left[80\;100\right],\;i=1\sim 4,\left(unit:\;mm\right)\;$;(2)$z_{i}\epsilon \left[43\;63\right],\;i=1,2,5,6,\left(unit:\;mm\right)\;$;(3)$z_{i}\epsilon \left[-51-31\right],\;i=3,4,7,8,\left(unit:\;mm\right)$.


### 4.3. Determination of Objective Function

The excitation sources of the frequency response function are set to the torsional excitations around the dither directions of no. 1~3 RLGs to simulate the mechanical dithers of RLGs (i.e., the angular accelerations of the dither cavities of RLGs around the *x*, *y*, and *z* axes of the centroid coordinate system in the Figure 5). The response directions are set to the six DOFs for each of the RLGs.

Under each RLG excitation, and in six DOFs of each RLG, only one response is in the same direction, and the other five responses are in different directions (hereinafter referred to as coupled frequency response), as shown in Table 2. $H_{xrx}$, $H_{yry}\;$ and $\;H_{zrz}\;$ in the table represent the responses in the same direction, and the rest are coupled frequency responses.

The optimization objective is set to minimize the ratio of the coupled frequency response to the same direction frequency response, that is, the following:(27)$$\{\begin{array}{c} Min\left(\frac{H_{xx}}{H_{xrx}}\right),Min\left(\frac{H_{xy}}{H_{xrx}}\right),Min\left(\frac{H_{xz}}{H_{xrx}}\right),Min\left(\frac{H_{xry}}{H_{xrx}}\right),Min\left(\frac{H_{xrz}}{H_{xrx}}\right),\\ Min\left(\frac{H_{yx}}{H_{yry}}\right),Min\left(\frac{H_{yy}}{H_{yry}}\right),Min\left(\frac{H_{yz}}{H_{yry}}\right),Min\left(\frac{H_{yrx}}{H_{yry}}\right),Min\left(\frac{H_{yrz}}{H_{yry}}\right),\\ Min\left(\frac{H_{zx}}{H_{zrz}}\right),Min\left(\frac{H_{zy}}{H_{zrz}}\right),Min\left(\frac{H_{zz}}{H_{zrz}}\right),Min\left(\frac{H_{zrx}}{H_{zrz}}\right),Min\left(\frac{H_{zry}}{H_{zrz}}\right), \end{array}$$

According to Equation (27), the final optimization objective is the sum of the above 15 single objectives, which is the following:(28)$$F=Min\left(\beta_{1}\underset{i=x,y,z}{\sum }\underset{j=x,y,z}{\sum }\frac{H_{ij}}{H_{iri}}+\beta_{2}\underset{\begin{array}{c} i=x,y,z,\\ i\neq j \end{array}}{\sum }\underset{j=x,y,z}{\sum }\frac{H_{irj}}{H_{iri}}\right),$$
where $\beta_{1}$ and $\beta_{2}$ are the weighting factors.

### 4.4. Optimization Results and Discussion

The optimized positions of each vibration isolator relative to the IMU centroid are given in Table 3, in which the bold figures are the optimal values.

The frequency response characteristics of optimized IMU structure and the initial structure are compared, as shown in Figure 7. It shows the 3 × 3 array of the frequency response curves. The diagonal and non-diagonal terms are the frequency response curves in the same directions and coupled directions, respectively. The blue curves are for the initial IMU, and the red ones are for the optimized IMU with the optimal positions of vibration isolators. The blue circles in the figure represent the coupled frequencies in each coupled frequency response curve.

According to the IMU structure configuration in Figure 5 and the results of the frequency response analyses in Figure 7, it can be seen that the coupled vibration of the system is well attenuated after optimizing the positions of the vibration isolators relative to the IMU centroid. In Figure 7, the figures in the first row are the responses under a torsional excitation along the *x* direction, and the resonant peaks of 570 Hz and 670 Hz are the coupled response peaks. The figures in the second row are the response under a torsional excitation along the *y* direction, and the resonant peaks of 490 Hz and 670 Hz are the coupled response peaks. The figures in the third row are the responses under a torsional excitation along the *z* direction, and the resonant peaks of 490 Hz and 570 Hz are the coupled response peaks.

The amplitudes of each of the coupled response peaks before and after the IMU structure optimization are shown in Table 4.

As shown in Table 4, the coupled vibration of the system based on the previous IMU design principle is further attenuated by the IMU decoupling optimization method proposed in this paper. The transmissibility of coupled vibration is decreased by 6 dB at least, and by 20 dB at most.

## 5. Experimental Verification of IMU Dynamic Decoupling Optimization Method

In this section, the vibration response tests and RLGs’ accuracy tests under the dither excitations of RLGs were done to demonstrate the IMU dynamic decoupling optimization method.

### 5.1. Testing Object

Two sets of sensor supporting structures are manufactured according to the above two design methods, respectively. One is based on the dynamic decoupling conditions of Equations (24) and (25), which is called as the IMU structure before optimization. The other is based on the decoupling optimization method proposed in this paper, which is named as the IMU structure after optimization. The distribution positions of the vibration isolators of the two sets of sensor supporting structures are given in Table 1 and Table 3, respectively, and the other physical properties are almost the same, which are shown in Table 5.

By selecting the same group of RLGs as inertial sensors and the same type of rubber dampers as vibration isolators, two sets of IMU prototypes are assembled as the test objects.

The nominal parameters of the selected three RLGs in the test are given in Table 6, where the accuracy of the RLGs is regarded as the standard deviation derived from the data sequence formed by the mean values of the measuring signals per 100 s in the length of 3 h.

### 5.2. Testing Scheme

In different IMU structures, the accuracy of three RLGs are tested; meanwhile, the coupled vibration responses of each IMU structure are obtained with the vibration test system. The corresponding measured accuracy of the RLGs and the vibration responses of each IMU structure are compared to verify the effectiveness of the IMU dynamic decoupling optimization proposed in this paper.

The same set of hardware modules to meet the normal operation requirements of each RLG in IMU are used in the test, including the IMU power module (supplying 15 V to each RLG) and signal processing module (resampling and outputting one datum per second for each RLG). Meanwhile, the B&K vibration test system made in Denmark is used to obtain the vibration response of the sensor supporting structure when three RLGs dither simultaneously. The main parameters of B&K vibration test system and the environment of the test are shown in Table 7.

The test configuration is shown in Figure 8 and the signal flow diagram in the test is shown in Figure 9.

The three-axis accelerometers are attached to the vertices of each sensor supporting structure in the IMU, and the serial number of each accelerometer is shown in Figure 10, as well as its coordinate system. Simultaneously, the axes of each accelerometer are respectively required to be along with the sensitive axes of the RLGs in IMU. In Figure 10, the blue arrows are the sensitive axes of the three RLGs in IMU, and the red arrows are the measuring axes of eight accelerometers. This layout method makes that each accelerometer could be easily sensitive to the linear acceleration owing to the dither of three RLGs in IMU. For example, the *x*-axis of no. 1 accelerometer is easily sensitive to the dither components of 490 Hz and 670 Hz, while the dither components of the frequency response of 570 Hz will be considered as coupled vibration. In this way, the frequency response of each measuring axis of the eight accelerometers includes two frequency components along with the excitation direction (hereinafter referred to as the same direction response), and one frequency component orthogonal to the excitation direction (that is, the coupled response), as shown in Table 8.

### 5.3. Testing Results

The acceleration responses are collected in the range of 0~800 Hz, and the frequency resolution is set to 0.5 Hz. The total 24-axis vibration frequency responses of eight accelerometers under three gyro dither excitations in each IMU are obtained. For the sake of the length of this paper, it only shows the measured vibration frequency response curves of the no. 1 and no. 8 accelerometers in the three axes, as shown in Figure 11. The results of the resonant peak amplitude at the coupled vibration frequency before and after IMU optimization are shown in Table 9.

The test results in Table 9 show that, after the dynamic decoupling optimization, the 24 groups of acceleration responses in IMU are effectively attenuated, among them, 20 groups are more than 50% attenuated, and the coupled vibration amplitude in the optimized IMU is attenuated by 93.4% compared with the original IMU at most. Therefore, compared with the previous IMU design principle, the coupled vibration could be further suppressed by the dynamic decoupling optimization method proposed in this paper. In the whole optimization process, the positions of the vibration isolators relative to the IMU centroid are only adjusted, which relaxes the strict requirements of the mass distribution of IMU in the previous design principles, such as referred in the works of [19,20,21]. Thus, the method proposed in this paper is more suitable for the practical engineering application.

The measured accuracy of the RLGs in IMU before and after optimization are compared, as shown in Figure 12. After three hours of testing, the accuracy of the RLGs after IMU optimization was improved compared with that before optimization. The reason is that the coupled vibration caused by the simultaneous dither of three RLGs is effectively attenuated after the IMU decoupling optimization, which makes each RLG work more stably in IMU.

The repeatability of the accuracy results of the RLGs in two sets of IMU structures is verified. Through five repeated tests, the comparative results of the measured accuracy of the three RLGs in two sets of IMU structures are shown in Table 10.

From Table 10, except for a few results are shown as small perturbations, most of the measured accuracy results are improved more than 1 × 10^−4^ °/h, which could be a good promotion for a long-endurance navigation. Through the above repeatability of tests, the effectiveness of the proposed method based on the dynamic modeling and decoupling design of IMU structure is verified.

## 6. Conclusions

In this paper, the multi-rigid body dynamic model of IMU system is modeled based on the topology analysis method subject to the constraint relations, and the multi-DOF coupled vibration behavior in IMU is clearly discussed under the RLGs’ dither excitation. Simultaneously, on the basis of the IMU dynamic model and focused on the vibration frequency response, the new dynamic decoupling method is proposed to optimize the sensor supporting structure of IMU by adjusting the positions of vibration isolators relative to the centroid of IMU. The testing results show that, compared with the previous optimal design principle through adjusting the mass distribution of IMU, the proposed dynamic decoupling optimization method could further suppress the coupled vibration and improve the measuring accuracy of the RLGs in the IMU system. This paper provides a more effective design idea for IMU to attenuate the coupled dither excitation of RLGs and ensure the accuracy of in IMU.

## Figures and Tables

**Figure 1 sensors-20-00111-f001:**
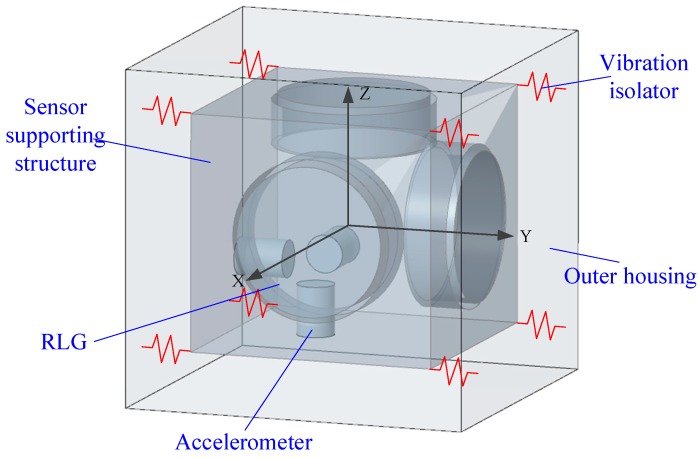
Structure diagram of laser gyro inertial measurement unit (IMU). RLG, ring laser gyro.

**Figure 2 sensors-20-00111-f002:**
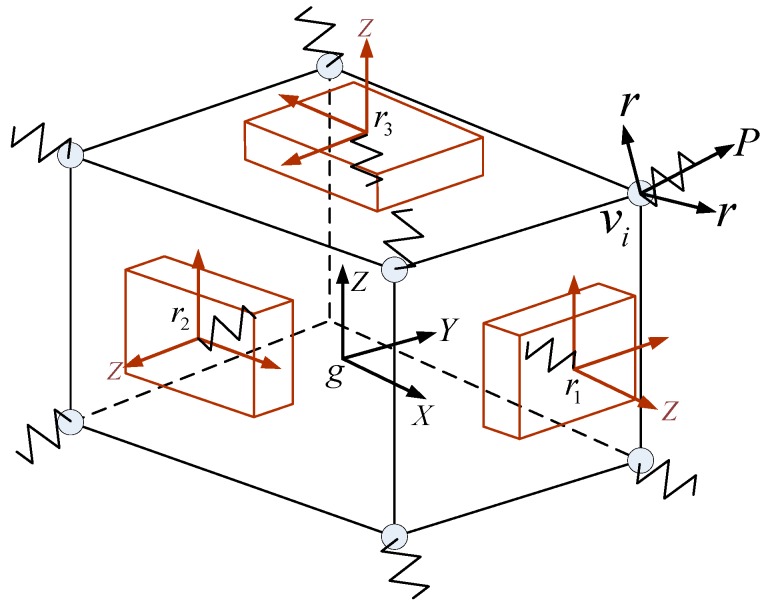
The schematic diagram of IMU dynamic model.

**Figure 3 sensors-20-00111-f003:**
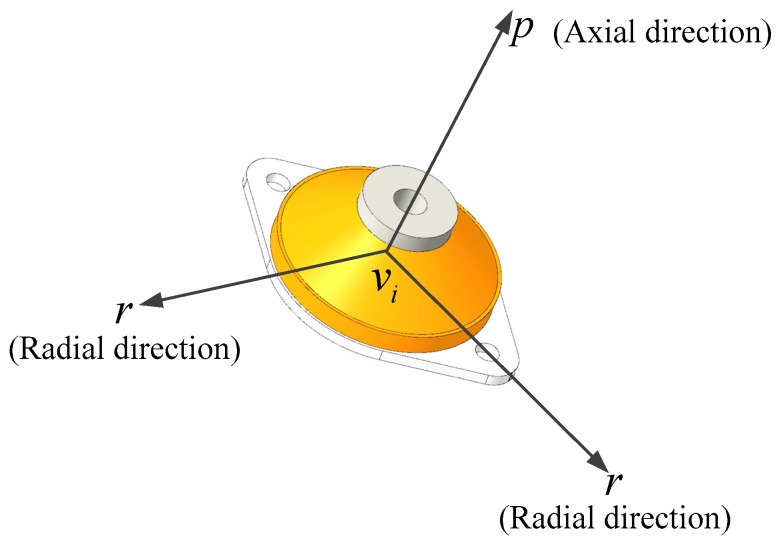
The structure of the vibration isolator.

**Figure 4 sensors-20-00111-f004:**
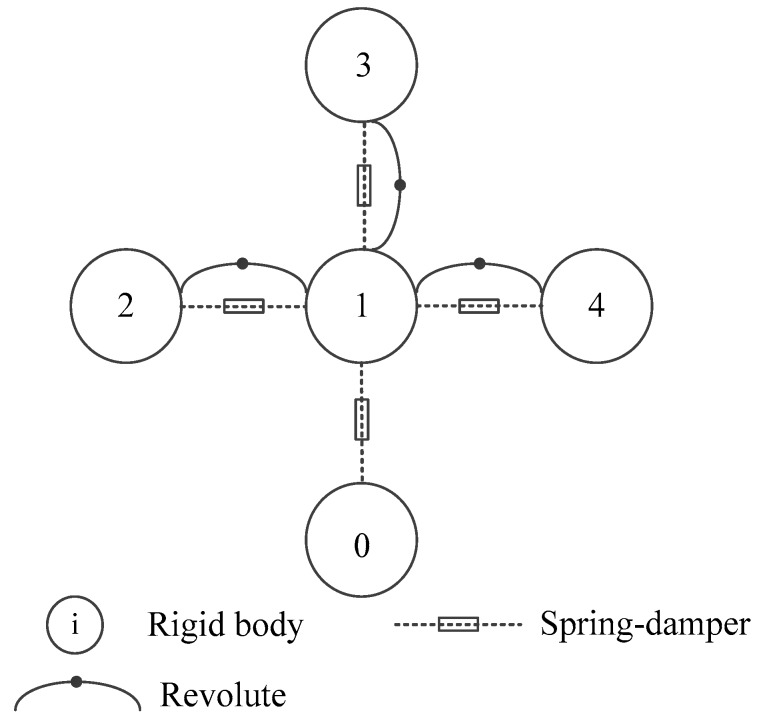
The constraint topological form of IMU.

**Figure 5 sensors-20-00111-f005:**
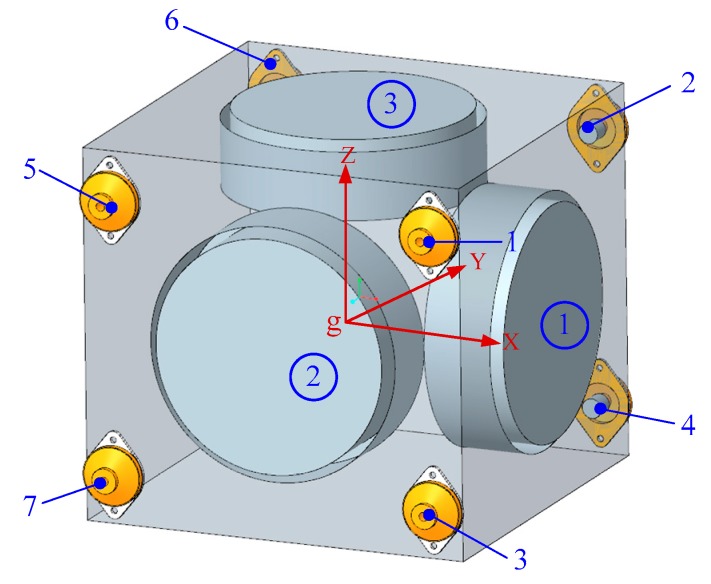
The IMU structure and the serial number of vibration isolators and RLGs.

**Figure 6 sensors-20-00111-f006:**
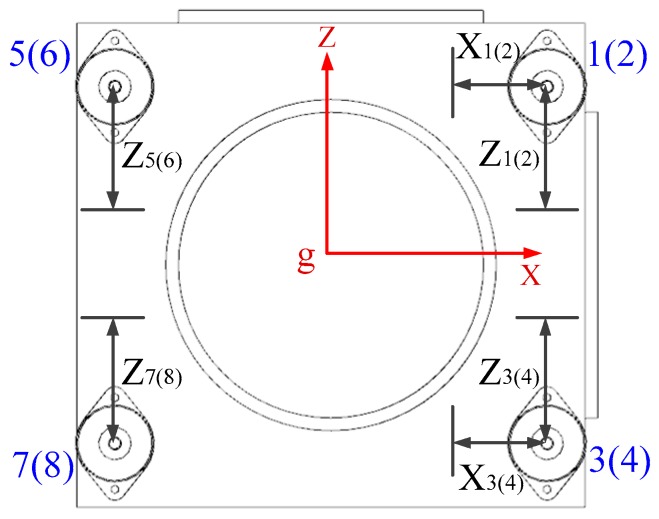
The optimization design parameters of IMU.

**Figure 7 sensors-20-00111-f007:**
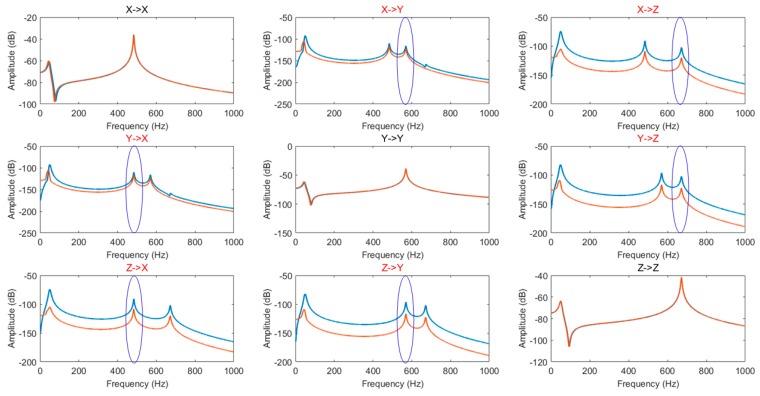
Comparison with the frequency response characteristics before and after IMU optimization.

**Figure 8 sensors-20-00111-f008:**
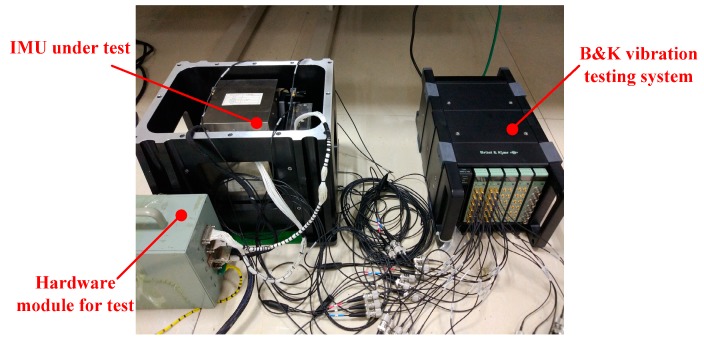
The configuration of the experiment.

**Figure 9 sensors-20-00111-f009:**
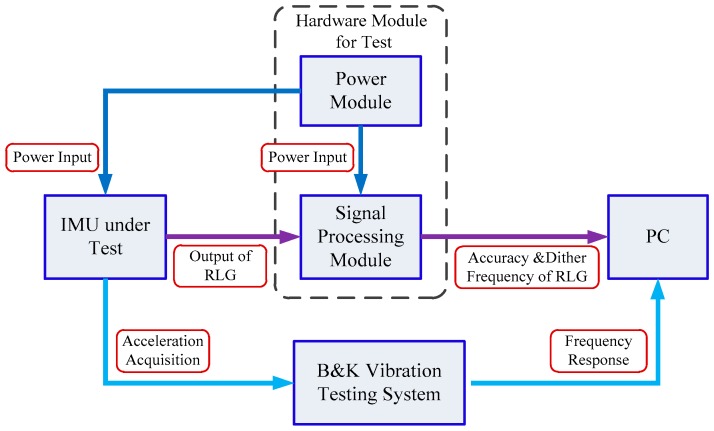
The signal flow diagram in the IMU test. (PC in the figure means Personal Computer)

**Figure 10 sensors-20-00111-f010:**
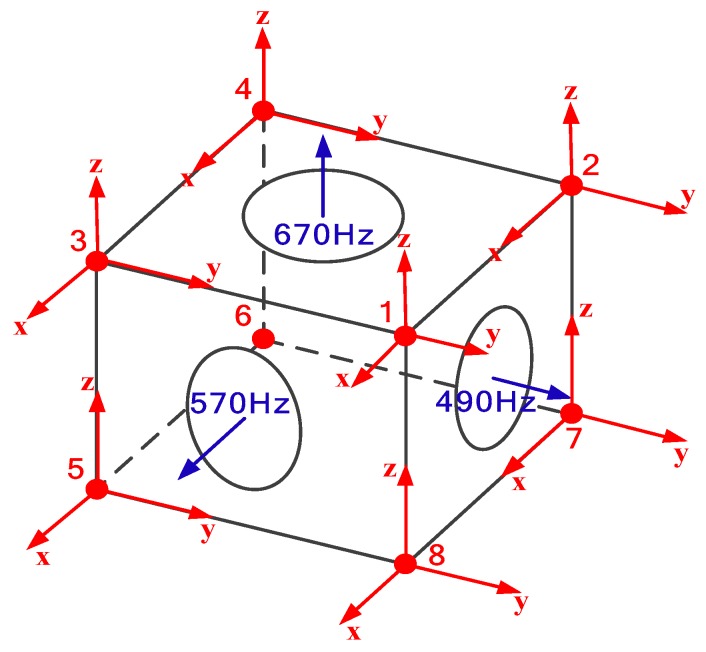
The layout of the measuring points of the three-axis accelerometers in the vibration test.

**Figure 11 sensors-20-00111-f011:**
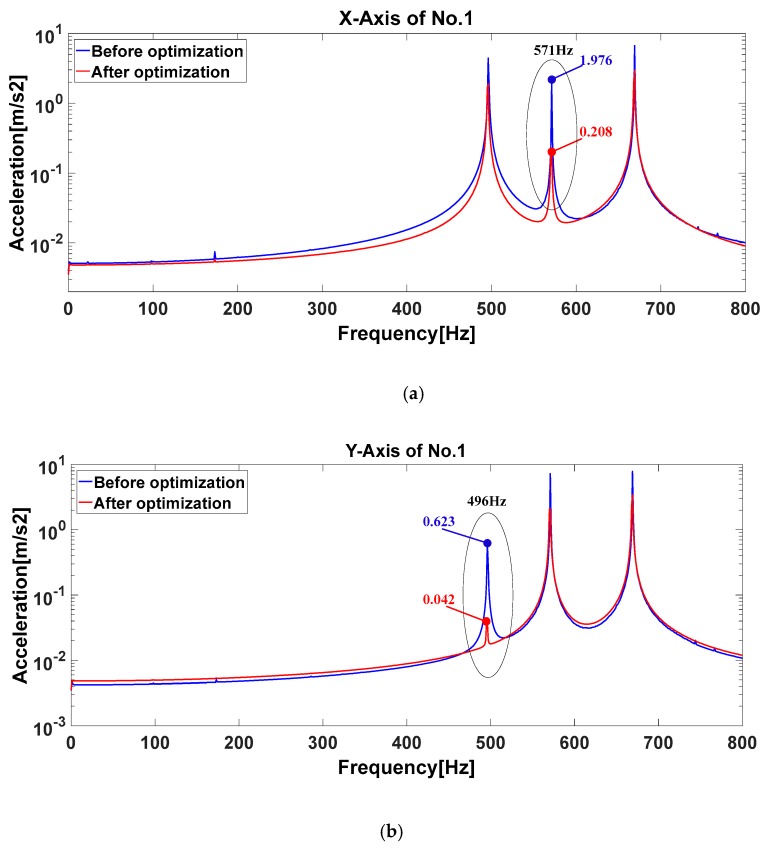
The comparisons of the measured coupled vibration responses of no. 1 and 8 accelerometers before and after IMU optimization. (**a**) The measured accelerations of *x*-axis of no. 1 accelerometer before and after optimization; (**b**) The measured accelerations of *y*-axis of no. 1 accelerometer before and after optimization; (**c**) The measured accelerations of *z*-axis of no. 1 accelerometer before and after optimization; (**d**) The measured accelerations of *x*-axis of no. 8 accelerometer before and after optimization; (**e**) The measured accelerations of *y*-axis of no. 8 accelerometer before and after optimization; (**f**) The measured accelerations of *z*-axis of no. 8 accelerometer before and after optimization.

**Figure 12 sensors-20-00111-f012:**
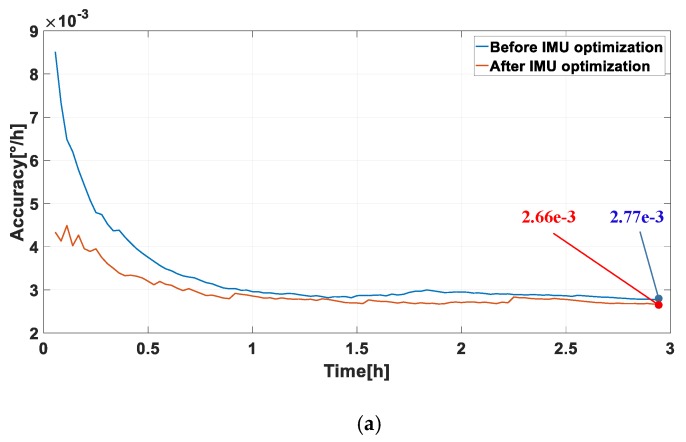
The comparisons of the measured accuracy of three RLGs in two sets of IMU structures. (**a**) The measured accuracy of gyro no. 1 with 490 Hz dither frequency; (**b**) the measured accuracy of gyro no. 2 with 570 Hz dither frequency; (**c**) the measured accuracy of gyro no. 3 with 670 Hz dither frequency.

**Table 1 sensors-20-00111-t001:** The initial positions of vibration isolators relative to the inertial measurement unit (IMU) centroid.

Serial Number	*x* (mm)	*y* (mm)	*z* (mm)
1	100.1	−87.6	62.8
2	100.1	82.4	62.8
3	100.1	−87.6	−51.2
4	100.1	82.4	−51.2
5	−88.9	−87.6	62.8
6	−88.9	82.4	62.8
7	−88.9	−87.6	−51.2
8	−88.9	82.4	−51.2

**Table 2 sensors-20-00111-t002:** The components of excitation and response in IMU.

Excitation Direction	Direction of Frequency Response					
	Translational Response			Revolutional Response		
	X	Y	Z	X	Y	Z
**Revolution Excitation of** X	$\;H_{xx}\;$	$\;H_{xy}\;$	$\;H_{xz}\;$	$H_{xrx}\;$	$H_{xry}\;$	$H_{xrz}$
**Revolution Excitation of** Y	$H_{yx}$	$H_{yy}$	$H_{yz}$	$H_{yrx}$	$H_{yry}$	$H_{yrz}\;$
**Revolution Excitation of** Z	$H_{zx}$	$H_{zy}\;$	$H_{zz}\;$	$H_{zrx}\;$	$H_{zry}\;$	$H_{zrz}\;$

**Table 3 sensors-20-00111-t003:** The optimized positions of the vibration isolators relative to the IMU centroid.

Serial Number	In *X* Direction (mm)	In *Y* Direction (mm)	In *Z* Direction (mm)
1	**93.3**	−87.6	**62.1**
2	**91.3**	82.4	**62.1**
3	**96.3**	−87.6	**−49.5**
4	**96.3**	82.4	**−46.5**
5	−88.9	−87.6	**62.6**
6	−88.9	82.4	**66.6**
7	−88.9	−87.6	**−47.5**
8	−88.9	82.4	**−50.5**

**Table 4 sensors-20-00111-t004:** Comparison with the coupled response peaks of IMU structures before and after optimization.

Situation of IMU Structure	Coupled Response Peaks in Each Coupled Direction (dB)					
	$H_{xry}\;$	$\;H_{xrz}\;$	$H_{yrx}\;$	$H_{yrz}\;$	$H_{zrx}\;$	$H_{zry}\;$
Before IMU optimization	−116	−102	−110	−102	−91	−97
After IMU optimization	−122	−120	−117	−122	−108	−117

**Table 5 sensors-20-00111-t005:** The physical properties of two sets of sensor supporting structures. RLG, ring laser gyro.

Physical Property	IMU Structure before Optimization	IMU Structure after Optimization
Material	Aluminum alloy	Aluminum alloy
Weight (kg)	2.89 kg	2.88 kg
Inertia (${\mathrm{kgmm}}^{2}$) (Including three RLGs)	$\left[\begin{array}{ccc} 6.56\times 10^{4} & -1.09\times 10^{2} & 6.61\times 10^{2}\\ -1.09\times 10^{2} & 7.75\times 10^{4} & -3.55\times 10^{2}\\ 6.61\times 10^{2} & -3.55\times 10^{2} & 7.54\times 10^{4} \end{array}\right]$	$\left[\begin{array}{ccc} 6.64\times 10^{4} & 1.24\times 10^{2} & 5.69\times 10^{2}\\ 1.24\times 10^{2} & 7.84\times 10^{4} & -2.09\times 10^{2}\\ 5.69\times 10^{2} & -2.09\times 10^{2} & 7.62\times 10^{4} \end{array}\right]$

**Table 6 sensors-20-00111-t006:** The nominal parameters of the three RLGs.

Nominal Parameters	RLG No. 1	RLG No. 2	RLG No. 3
Bias drift	0.075 °/h	0.11 °/h	0.106 °/h
Angle random walk	$0.00065\;{}^{{}^{\circ}}/\surd h\;$	$0.00057\;{}^{{}^{\circ}}/\surd h\;$	$0.00041\;{}^{{}^{\circ}}/\surd h\;$
Accuracy	0.0026 °/h	0.0027 °/h	0.0019 °/h
Dither frequency	493 Hz	568 Hz	667 Hz

**Table 7 sensors-20-00111-t007:** The main parameters of B&K system and the environment of the test.

**Parameters of** **B&K System**	**Analyzing Bandwidth**	DC-51.2 kHz
	**Dynamic Range**	160 dB
	**Input Voltage Range**	±10 V
	**A/D**	24 bit
	**Sensitivity of Acceleration**	10 mV/g
**Environment** **of Test**	**Location**	30.392 °N, 114.349 °E
	**Temperature**	25 ± 2 °C

**Table 8 sensors-20-00111-t008:** Corresponding relations of the response components of each measuring point of each accelerometer.

Measuring Points	Measuring Axis	Same Direction Response Frequency Components (Hz)		Coupled Response Frequency Components (Hz)
**1~8**	*x*	490	670	570
	*y*	570	670	490
	*z*	490	570	670

**Table 9 sensors-20-00111-t009:** The comparison result of the dynamic decoupling effect before and after IMU optimization.

Number of Measuring Point	Direction	Resonant Peak Amplitude (m/s^2^)		Reduction Ratio of Amplitude
		Before IMU Optimized	After IMU Optimized	
1	*X*-axis	1.976	0.208	89.5%
	*Y*-axis	0.623	0.041	93.4%
	*Z*-axis	0.846	0.338	54.1%
2	*X*-axis	0.201	0.175	12.9%
	*Y*-axis	0.305	0.104	65.9%
	*Z*-axis	0.704	0.326	53.7%
3	*X*-axis	0.631	0.354	43.9%
	*Y*-axis	0.754	0.223	70.4%
	*Z*-axis	0.451	0.139	69.2%
4	*X*-axis	0.655	0.318	51.5%
	*Y*-axis	0.189	0.047	75.1%
	*Z*-axis	0.594	0.24	59.6%
5	*X*-axis	0.571	0.168	70.6%
	*Y*-axis	5.631	2.702	52%
	*Z*-axis	0.455	0.254	44.2%
6	*X*-axis	0.268	0.058	78.4%
	*Y*-axis	0.539	0.14	74%
	*Z*-axis	0.318	0.208	34.6%
7	*X*-axis	1.034	0.166	83.9%
	*Y*-axis	0.463	0.101	78.2%
	*Z*-axis	2.069	0.649	68.6%
8	*X*-axis	1.455	0.176	87.9%
	*Y*-axis	0.433	0.111	74.4%
	*Z*-axis	1.839	0.507	72.4%

**Table 10 sensors-20-00111-t010:** The repeatability of the measured accuracy of the three RLGs in two sets of IMU structures.

Times of Tests	Serial Number of RLG	Measured Accuracy (°/h)		Trend of Accuracy
		Before IMU Optimized	After IMU Optimized	
1	1	2.77 × 10^−3^	2.66 × 10^−3^	Improvement
	2	2.83 × 10^−3^	2.81 × 10^−3^	Small perturbation
	3	2.34 × 10^−3^	1.94 × 10^−3^	Improvement
2	1	2.84 × 10^−3^	2.65 × 10^−3^	Improvement
	2	2.91 × 10^−3^	2.93 × 10^−3^	Small perturbation
	3	2.31 × 10^−3^	1.99 × 10^−3^	Improvement
3	1	2.71 × 10^−3^	2.54 × 10^−3^	Improvement
	2	2.81 × 10^−3^	2.69 × 10^−3^	Improvement
	3	2.11 × 10^−3^	2.01 × 10^−3^	Improvement
4	1	2.83 × 10^−3^	2.68 × 10^−3^	Improvement
	2	2.72 × 10^−3^	2.75 × 10^−3^	Small perturbation
	3	2.15 × 10^−3^	1.98 × 10^−3^	Improvement
5	1	2.81 × 10^−3^	2.69 × 10^−3^	Improvement
	2	2.88 × 10^−3^	2.75 × 10^−3^	Improvement
	3	2.21 × 10^−3^	1.93 × 10^−3^	Improvement

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
