# Peer review of "Coupled Dynamic Analysis and Decoupling Optimization Method of the Laser Gyro Inertial Measurement Unit"

_sensors, 2019, doi:10.3390/s20010111_

Round 1

Reviewer 1 Report

The paper reports on interesting experimental results on a decoupling optimization method of RLG-based IMUs.

The paper topic is interesting and the manuscript is clear.

The discussion of the state of the art ignores the RLG competing technologies: FOG, HRG, and RMOG. These technologies should be cited. See DOI: 10.2971/jeos.2014.14013

The sub-section “5.2. Testing scheme” should include additional details on the setup, i.e. IMU under test and B&K vibration testing system. The bias drift and ARW of the RLG under test should be mentioned.

The repeatability of experimental results is impossible. This is a serious issue of the manuscript.

I suggest a major revision.

Author Response

Reviewer #1

Response to comment: The discussion of the state of the art ignores the RLG competing technologies: FOG, HRG, and RMOG. These technologies should be cited. See DOI: 10.2971/jeos.2014.14013.

Reply: Thank the Reviewer for this comment. Though this manuscript focuses on the optimal design of the IMU structure with regard to RLGs, it is undeniable that except RLGs, there are various kinds of gyroscopes being researched or applied in the fields of inertial navigation, such as FOGs, HRGs, and RMOGs, etc. Therefore, as the Reviewer’s good comment, the revised manuscript gives an outline of the above types of gyros in the section of the introduction. We list the detail here: “Gyroscopes are the key angular rate sensors. In the current applications of inertial navigation, two types of gyroscopes are dominated, i.e. the optoelectronic types including Ring Laser Gyros (RLGs) [1,2] and Fiber Optic Gyros (FOGs) [3,4] which are based on the Sagnac effect, and solid vibratory types such as Hemispherical Resonator Gyros (HRGs) [5,6] which are based on the Coriolis force. Among these products, The RLG is the main inertial device due to the high performance and reliabilities to vibration and temperature, especially for the harsh environments. The FOG senses the angular rate by phase shift between two counter-propagating waves in a fiber coil, and its performance is up to a similar grade to that of the RLG, but it may be more easily affected by external dynamic, thermal or stress perturbations [7]. The HRG detects the precession angle of the stress wave in a quartz hemispherical resonator so as to sense the angular rate [8]. Compared to the optoelectronic gyros, the HRG has superiority for its high reliability, low power, and good environmental adaptability. On the other hand, the chip-scale optoelectronic gyroscopes will play important roles in the gyroscope market due to the low costs and miniaturization [9]. The Resonant Micro-Optic Gyro (RMOG) is the promising sensor in the field of integrated optical gyros, and more innovative researches will aim to RMOG’s key technology, i.e. the silica-on-silicon ring resonator with ultra-high Q-factor and the monolithic integration based on gyro on a chip [10,11].

Even though more and more types of high-performance gyroscopes are researched and manufactured, the RLG  still plays an extremely important role in the field of high precision inertial navigation due to the mature manufacturing process. In the family of RLG, the mechanical dithered RLG is most widely applied, which overcomes the gyro's lock-in effect by the dither bias system and ensures the high measurement accuracy of the gyro for low angular rate [12].”

Response to comment: The sub-section “5.2. Testing scheme” should include additional details on the setup, i.e. IMU under test and B&K vibration testing system. The bias drift and ARW of the RLG under test should be mentioned.

Reply: Thank the Reviewer very much for pointing this out. The authors add the additional texts on the details of the two sensor supporting structures in IMUs compared, the properties of three RLGs used in the tests such as bias drifts and ARWs, and also the information of B&K vibration testing system as well as the test environment in the sub-sections of “5.1. Testing object” and “5.2. Testing scheme”, respectively.

Response to comment: The repeatability of experimental results is impossible. This is a serious issue of the manuscript.

Reply: Thank the Reviewer very much, and this is indeed a serious issue. The authors test the RLGs’ accuracy in the comparative sets of IMU structures 5 times repeatedly and add the corresponding statements in the sub-section of “5.3. Testing results” in the revised manuscript. A good consistency has shown in the repeatability test, and the effectiveness of the proposed set of methods including dynamic modeling and decoupling design of IMU structure can be fully verified.

Reviewer 2 Report

A new IMU dynamic decoupling optimization method is proposed to minimize the coupled vibration frequency response. This method is innovative, but there are still many problems.

English language and style must be improved. References are insufficient. The proposed method is compared with the original design principle, which can't fully prove the effectiveness of the proposed method. The authors need to add comparative experiments of other optimization methods.

Author Response

Reviewer #2

Response to comment: The English language and style must be improved.

Reply: Thank the Reviewer for this reminder. The authors have reviewed the whole manuscript and tried our best to correct the mistakes in the written expressions (marked in blue in the revised manuscript).

Response to comment: References are insufficient

Reply: Thank the Reviewer for this comment. The authors have added some literature as follows:

Theory of a novel high sensitivity optical fiber gyroscope Fundamentals of the interferometric fiber-optic gyroscope Solid-state wave gyroscope: a new generation inertial sensor [C]. The 24th Saint Petersburg International Conference on Integrated Navigation System, 2017 Lynch, D.; Hemispherical Resonator Gyro. IEEE Transactions on Aerospace and Electronic System 1984 Culshaw, “The optical fiber Sagnac interferometer: an overview of its principles and applications,” Meas. Sci. Technol. 17, R1–R16 (2006). A New Control Mode Greatly Improving Performance of Axisymmetrical Vibrating Gyroscope Liu, W. Zhang, W. Chen, K. Li, F. Dai, F. Cui, X. Wu, et al., “The development of micro-gyroscope technology,” J. Micromech. Microeng. 19, 113001 (2009). Recent advances in miniaturized optical gyroscopes System test of an optoelectronic gyroscope based on a high Q-factor InP ring resonator. J.Ewins; Modal Testing, Theory Practice and Application, Second Edition

Response to comment: The proposed method is compared with the original design principle, which can't fully prove the effectiveness of the proposed method. The authors need to add comparative experiments of the other optimization methods.

Reply: Thank the Reviewer for this comment. We are very sorry for the misunderstanding of “the original design principle”. Actually, the statement of “the original design principle” in the manuscript refers to “the optimal design principle in the previous research”, the typical instances of which are the conclusions in references [19] to [21], i.e. Eqs (16) to (18) in [19], or the section 3.1 in [21]. For easy reference to the Reviewer, we list the corresponding texts as follows:

In “[19]. Isotropic Design Method of Suspension System of Dithered RLG Strapdown Inertial Measurement Unit”:

In “[21]. The optimized design method of vibration isolation system in mechanically dithered RLG POS based on motion decoupling”:

The above optimal design principles in the references are put forward by dynamic modeling and analysis of laser gyro IMU, the research of which is important to the design of laser gyro IMU. However, there are still some flaws in the previous research on the IMU optimal design, i.e., the condition of “” is always hard to satisfy in the actual IMU design. On the other hand, in order to further eliminate the coupled vibration due to dither of RLG, the optimization of the frequency response of the IMU structure may be considered as the improvement method, and this is the pointcut of our manuscript.

The comparative objects in experiments in the manuscript are set as the sensor supporting structures made on the bases of the proposed optimization method and the previous method respectively. Therefore, the comparative experiments between the optimization method in our manuscript and the so-called “original design principle” are actually the comparisons between the whole set of methods (including dynamic modelling applied for frequency analysis, and decoupling design of IMU structure) in our manuscript and the previous optimization methods (such as the Newton-Eulerian modelling and the diagonalizing of the mass and stiffness matrices). That is to say, the results of the comparative experiments can verify the effectiveness of the proposed optimization method superior to the optimal design principle in the previous research such as [19] to [21].

We want to apologize for the mistake of expression of “original design principle” again, and modify expression of “original design principle” to that of “previous optimal design principle” in the revised manuscript, meanwhile, add the specific texts on the “previous optimal design principle” in section 3.2: “The similar decoupling conditions can be found in the references [19-21], such as Eqs. (16) to (18) in [19], or the section 3.1 in [21].”.

At the end of this response, in order to clearly point out the comparisons between the proposed method and the previous method as far as the authors’ understanding, the schematic diagram is summarized as follows:

Finally, we appreciate for Editors/Reviewers’ warm work earnestly and hope that the correction will meet with approval.

Round 2

Reviewer 1 Report

I suggest the manuscript publication in the present form.

Author Response

Thank you very much for your comments.

Reviewer 2 Report

The author's article has been greatly improved, but there are still two problems.

The investigation of the new method is relatively inadequate. The optimal design principle in the reference [19] is used to compare with the proposed method. But adding newer method is more effective, such as methods in references [23][24] .

Author Response

Response to Review Comments

Thanks to the Editor for coordinating the review process and the Reviewer for the thoughtful comments on our manuscript. We have seriously considered the comments of the Reviewer, and revised the relevant parts of the manuscript. Revisions are marked in blue based on the previous version of the manuscript, and responses to the comments of the Reviewer are listed in the following.

Response to comment: The investigation of the new method is relatively inadequate.

Reply: Thanks to the Reviewer for this comment. To the new modelling and optimization method proposed in the manuscript, the authors’ understandings are as follows:

The previous researches (Refs.[18-25] in the revised manuscript) on the dynamic design of laser gyro IMU always focus on two categories, specifically, the calibration of the IMU (Refs. [18-21]) and the optimal design of IMU structure (Refs. [23-25]). Since the errors such as biases or misalignments of RLGs in IMU are always represented as the constants or time-varying variables, the time-domain based dynamic analysis methods are appropriate and effective. So it is in the previous optimal design of IMU structure by diagonalizing of the system stiffness and the mass matrices. Despite it is easy to be realized by diagonalizing of the system stiffness and the mass matrices, vibration decoupling of laser gyro IMU due to the dithers of RLGs is actually a problem about frequency-domain analysis and optimization, since the dither of each RLG in the IMU behaves in different frequency bands. Vibration decoupling exactly means the minimum of the vibration in coupled direction, therefore, the frequency-domain-analysis based vibration decoupling method is very essential to the vibration decoupling in IMU (The conclusions of this manuscript could also verify this viewpoint). Till now, the frequency-domain-analysis based vibration decoupling method is still not applied in the research on the dynamic design of laser gyro IMU in spite of the wide applications in other engineering field So the complete set of modelling and optimization methods based on the frequency analysis is an important new idea of this manuscript, and could be a new perspective for the optimal design of the laser gyro IMU.

Therefore, according to the Reviewer’s suggestion, the authors have specifically revised the relevant texts in Introduction of the manuscript, and also added the relevant references, so as to present the whole idea of the manuscript. The revised parts about the investigation of the proposed vibration decoupling method are listed as follows: (Please see the revised manuscript marked blue for more details)

“Therefore, frequency-domain-analysis based modelling and optimization methods[26] may become the new key techniques for the vibration decoupling and optimization of the laser gyro IMU. Frequency-domain analytical methods focus on the analysis and optimization of the modal and frequency response for the mechatronic systems, which have been widely used in multiple precision manufacturing fields, i.e. the structural optimization of high-performance machine tools[27], the design of the key device for space applications[28], and NVH(Noise, Vibration and Harshness) analysis of the vehicle[29], etc.”

Response to comment: The optimal design principle in reference [19] is used to compare with the proposed method. But adding a newer method is more effective, such as methods in references [23][24].

Reply: Thanks to the Reviewer very much for this suggestion. Refs.[23] and [24] (The serial numbers are modified to [20] and [21] in the revised version) are the recent and valuable researches on the design of laser gyro IMU or INS. As mentioned in the introduction of the revised manuscript, Refs.[23] and [24] are both about the aspect of the calibration of the systematic error in IMU caused by the damper deformation.

Ref.[23] proposed a new calibration method of the g-sensitive misalignments of an RLG, and its important innovative point is the calibration involving the linear vibration environment, but the errors calibrated are the bias drifts of the RLGs in IMU, which always present the constants. The relevant text in Ref. [23] is as follows:

The modelling method used in Ref. [23] is the scope of the multi-degree-of-freedom kinematics, which is different from the dynamic modelling and optimization in our manuscript.

Ref.[24] proposed a novel dynamic analysis and experimental verification method for the RLGs’ deviation angles in dual-axis RINS. The dynamic solutions of the deviation angles are derived through dynamic analysis of the IMU, so the errors calibrated in Ref.[24] could be time-varying.

By comparing the dynamic modelling in Ref.[24] and in this manuscript, some differences are summarized and listed below:

Ref.[24]

This manuscript

Dynamic modelling method

l Newton-Euler and Runge-Kutta methods (Section 3.2)

l Convenient for time-domain solution

l Analytical mechanics and frequency analysis methods

l Convenient for frequency-domain solution

Decoupling analysis method

l Modal decoupling method (Section 4.2.1)

l Decoupling in a modal coordinate system

l Not used for the reduction of coupled vibration

l Vibration decoupling optimization

l Vibration decoupling in a physical coordinate system

Other differences

l The stiffnesses of dampers are defined as three-dimensional equality principle (Section 3.1)

l The different values of stiffnesses in each dimension of dampers can be considered in the modelling

To sum up, due to the different purpose from that in this manuscript, the decoupling analysis method in Ref.[24] is not used for the vibration decoupling optimization of the IMU structure, but for the time-domain solutions of the deviation angles of RLGs.

According to the Reviewer’s suggestion, the authors have added the relevant text about the comparisons with the Ref. [24] in section 3.2 of the revised manuscript, so as to show the effectiveness of the proposed method more definitely. The relevant parts are listed as follows: (Please see the revised manuscript marked blue for more details)

“The similar decoupling researches can be found in the references [21] and [23-25], such as the section 4.2.1 in [21], Eqs. (16) to (18) in [23], and the section 3.1 in [25],. Among the above previous researches, reference [21] is the recent and valuable study on the dynamic analysis of the laser gyro IMU, and it focuses on the calibration of the gyro deviation angle in the biaxial INS. Despite the different purpose of this paper, it uses the modal decoupling method to analyze the response of the gyro deviation angle. However, that decoupling in [21] is a mathematical decoupling in the modal coordinate system (not in an actual physical coordinate system), which is just applied for the subsequent time-domain analysis. ”

Finally, we appreciate for Editors/Reviewers’ warm work earnestly and hope that the correction will meet with approval.

Round 3

Reviewer 2 Report

The authors answered the questions seriously, but they can be improved further.

Aiming at the first comment, the authors added some relevant texts and references. But these methods were basically researched ten years ago, and they are not new. The authors should pay more attention to the research in recent years. For the second comment, the reviewer pointed out that "adding a newer method is more effective, such as methods in references[23][24]". The authors analyzed that method in Ref. [24] is not suitable for comparison with the proposed method. But the authors should choose another suitable and new method.

Author Response

Response to Review Comments

Thanks to the Editor for coordinating the review process and the Reviewer for the thoughtful comments on our manuscript. We have seriously considered the comments of the Reviewer, and revised the relevant parts of the manuscript. Revisions are marked in blue based on the previous version of the manuscript, and responses to the comments of the Reviewer are listed in the following.

Response to comment: Aiming at the first comment, the authors added some relevant texts and references. But these methods were basically researched ten years ago, and they are not new. The authors should pay more attention to research in recent years.

Reply: Thanks to the Reviewer for this comment. We update the relative researches in recent years in the revised manuscript, and list them as follows:

Stepan G , Kiss A K , Ghalamchi B , et al. Chatter avoidance in cutting highly flexible workpieces[J]. CIRP Annals - Manufacturing Technology, 2017. Mohanty S , Dwivedy S K . Nonlinear dynamics of piezoelectric-based active nonlinear vibration absorber using time delay acceleration feedback[J]. Nonlinear Dynamics, 2019(5). Haris A , Motato E , Mohammadpour M , et al. On the effect of multiple parallel nonlinear absorbers in palliation of torsional response of automotive drivetrain[J]. International Journal of Non-Linear Mechanics, 2017, 96.

Among the above recent researches, Ref. [28] employed the method of frequency response analysis to represent the regenerative chatter in cutting highly flexible workpiece so as to improve the robust stability in turning. In Ref. [29], a classical nonlinear dynamic analysis method such as harmonic balance method was used to derive the system model of the piezoelectric-based active vibration isolator, and a large amount of the frequency response analyses were involved. Meanwhile, NVH(Noise, Vibration and Harshness) analysis was always one of the most important tests in the modern automotive industry, and the relative researches based on frequency analysis could be found until recently, such as Ref. [30].

Response to comment: The authors analyzed that method in Ref. [24] (The serial number is modified to [21] in the revised version) is not suitable for comparison with the proposed method. But the authors should choose another suitable and new method.

Reply: Thanks to the Reviewer very much for this comment. As suggested by the Reviewer, we have tried our best to compare to the suitable researches in similar applications in the last 5 years, but unfortunately, so far, the specific decoupling optimization principle and relevant modelling method could only be tracked back to the researches in Ref. [25] in 2014. The comparisons between the method in Ref. [25] and that in our manuscript have been discussed focally in the first-revised manuscript.

Even with the limited new results on the vibration decoupling optimization of the dithered laser gyro IMU in the recent 5 years, the high importance of this study is still remained. It is because that more serious problems of the coupled vibration in new types of laser gyro IMUs may arise under the trends of the development of high reliability or miniaturization, i.e, the vibration will be increased with the decrease of the weight and volume of IMU structure.

Therefore, a relative recent research on a new type of skewed redundant IMU in Ref. [26] published in 2016 is added and discussed in the revised manuscript, which employs four RLGs to form a skewed redundant structure. In that configuration, the orthogonal layout of RLGs is violated and coupled vibration could be increased. In Ref. [26], the methods to reduce the coupled dither of RLGs was mentioned as follows:

As shown in the above texts in Ref. [26], the increased weight and volume of structure, and reasonable allocating of gyro dither frequencies are employed to reduce the coupling vibration, but no dynamic modelling and optimal design method are considered. In order to realize the vibration decoupling in that non-orthogonal structure of IMU, the frequency analysis and optimization are essential, and this is the scope of our manuscript.

Therefore, in spite of the lack of the new researches on the vibration decoupling optimization of IMU in last 5 years, we add the texts on Ref. [26] in the revised manuscript, so as to track the most recent research on the dynamic design of laser gyro IMU, and also to further highlight the referential value of the proposed method of vibration decoupling in laser gyro IMU due to the helplessness of the previous results in Refs. [23-25] for some non-orthogonal new types of laser gyro IMUs.

(Please inspect the modified details marked blue in Introduction and Section 3.2 of the revised manuscript)

We have tried our best to reply the Reviewer’s question. Meanwhile, the proposed method has been compared to the most recent methods in Refs. [23-25] published between 2013 and 2014, and also been emphasized by discussing the most recent and similar-scoped research in Ref. [26] published in 2016. We hope the Reviewer could weigh this response and the current states of relevant researches earnestly.
